# VISION-BASED DISCOVERY OF NONLINEAR DYNAMICS FOR 3D MOVING TARGET

## ABSTRACT

Data-driven discovery of governing equations has kindled significant interests in many science and engineering areas. Existing studies primarily focus on uncovering equations that govern nonlinear dynamics based on direct measurement of the system states (e.g., trajectories). Limited efforts have been placed on distilling governing laws of dynamics directly from videos for moving targets in a 3D space. To this end, we propose a vision-based approach to automatically uncover governing equations of nonlinear dynamics for 3D moving targets via raw videos recorded by a set of cameras. The approach is composed of three key blocks: (1) a target tracking module that extracts plane pixel motions of the moving target in each video, (2) a Rodrigues' rotation formula-based coordinate transformation learning module that reconstructs the 3D coordinates with respect to a predefined reference point, and (3) a spline-enhanced library-based sparse regressor that uncovers the underlying governing law of dynamics. This framework is capable of effectively handling the challenges associated with measurement data, e.g., noise in the video, imprecise tracking of the target that causes data missing, etc. The efficacy of the proposed method has been demonstrated through multiple sets of synthetic videos considering different nonlinear dynamics.

## 1 INTRODUCTION

Nonlinear dynamics is ubiquitous in nature. Data-driven discovery of underlying laws or equations that govern complex dynamics has drawn great attention in many science and engineering areas such as astrophysics, aerospace science, biomedicine, etc. Existing studies primarily focus on uncovering governing equations based on direct measurement of the system states, e.g., trajectory time series, (Bongard & Lipson, 2007; Schmidt & Lipson, 2009; Brunton et al., 2016; Rudy et al., 2017; Chen et al., 2021b; Sun et al., 2021). Limited efforts have been placed on distilling governing laws of dynamics directly from videos for moving targets in a 3D space, which represents a novel and interdisciplinary research domain. This challenge calls for a solution of fusing various techniques, including computer stereo vision, visual object tracking, and symbolic discovery of equations.

Let us consider a moving object in a 3D space. The scene of this object is recorded by a set of horizontally positioned, calibrated cameras at different locations. Discovery of the governing equations for the moving target first requires accurate estimation of its 3D trajectory directly from the videos, which can be realized based on computer stereo vision and object tracking techniques. Computer stereo vision, which aims to reconstruct 3D coordinates for depth estimation of a given target, has shown immense potential in the fields of robotics (Nalpantidis & Gasteratos, 2011; Li et al., 2021), autonomous driving (Ma et al., 2019; Peng et al., 2020), among others. Disparity estimation is a crucial step in stereo vision, as it computes the distance information of objects in a 3D space, thereby enabling accurate perception and understanding of the environment. Recent advances of deep learning has kindled successful techniques for visual object tracking e.g., DeepSORT (Wojke et al., 2017) and YOLO (Redmon et al., 2016). The aforementioned techniques lay a critical foundation to accurately estimate the 3D trajectory of a moving target for distilling governing equations, simply based on videos recorded by multiple cameras in a complex scene.

We assume that the nonlinear dynamics of a moving target can be described by a set of ordinary differential equations, e.g., $d\mathbf{y}/dt = \mathcal{F}(\mathbf{y})$, where $\mathcal{F}$ is a nonlinear function of the $d$-dimensional system state $\mathbf{y} = \{y_1(t), y_2(t), \ldots, y_d(t)\} \in \mathbb{R}^d$. The objective of equation discovery is to identify

the closed-form of $\mathcal{F}$ given observations of $\mathbf{y}$. This could be achieved via symbolic regression (Bongard & Lipson, 2007; Schmidt & Lipson, 2009; Sahoo et al., 2018; Petersen et al., 2021; Mundhenk et al., 2021; Sun et al., 2023) or sparse regression (Brunton et al., 2016; Rudy et al., 2017; Rao et al., 2023). When the measurement data is noisy and sparse, differentiable models (e.g., neural networks (Chen et al., 2021b), cubic splines (Sun et al., 2021; 2022)) are employed to reconstruct the system states, thereby forming the physics-informed learning scheme for more robust equation discovery.

Recently, attempts have been made toward scene understanding and prediction grounding physical concepts (Jaques et al., 2020; Chen et al., 2021a). Although a number of efforts have been placed on distilling the unknown governing laws of dynamics from videos for moving targets (Champion et al., 2019; Udrescu & Tegmark, 2021; Luan et al., 2022), the system dynamics was assumed in plane (e.g., in a 2D space). To our knowledge, distilling governing equations for a moving object in a 3D space (e.g., $d = 3$) directly from raw videos remains scant in literature. To this end, we introduce a unified vision-based approach to automatically uncover governing equations of nonlinear dynamics for a moving target in a predefined reference coordinate system, based on raw video data recorded by a set of horizontally positioned, calibrated cameras at different locations.

**Contributions.** The proposed approach is composed of three key blocks: (1) a target tracking module based on YOLO-v8 that extracts plane pixel motions of the moving target in each video data; (2) a coordinate transformation model based on Rodrigues' rotation formula, which allows the conversion of pixel coordinates obtained through target tracking to 3D spatial/physical coordinates in a predefined reference coordinate system given the calibrated parameters of only one camera; and (3) a spline-enhanced library-based sparse regressor that uncovers a parsimonious form of the underlying governing equations for the nonlinear dynamics. Through the integration of these components, it becomes possible to extract spatiotemporal information of a moving target from 2D video data and subsequently uncover the underlying governing law of dynamics. This integrated framework excels in effectively addressing challenges associated with measurement noise and data gaps induced by imprecise target tracking. Results from extensive experiments demonstrate the efficacy of the proposed method. This endeavor offers a novel perspective for understanding the complex dynamics of moving targets in a 3D space.

## 2 RELATED WORK

**Computer stereo vision.** Multi-view stereo aims to reconstruct a 3D model of the observed scene from images with different viewpoints (Schönberger et al., 2016; Galliani et al., 2016), assuming the intrinsic and extrinsic camera parameters are known.Recently, many endeavors have been based on deep learning to tackle this challenge, such as convolutional neural networks (Flynn et al., 2016; Huang et al., 2018) and adaptive modulation network with co-teaching strategy (Wang et al., 2021).

**Target tracking.** Methods for vision-based target tracking can be broadly categorized into two main classes: correlation filtering and deep learning. Compared to traditional algorithms, correlation filtering-based approaches offer faster target tracking (Mueller et al., 2017), while deep learning-based methods (Ciaparrone et al., 2020; Marvasti-Zadeh et al., 2021) provide higher precision.

**Governing equation discovery.** Data-driven discovery of governing equations can be realized through a number of symbolic/sparse regression techniques. The most popular symbolic regression methods include genetic programming (Koza, 1994; Bongard & Lipson, 2007; Schmidt & Lipson, 2009), symbolic neural networks (Sahoo et al., 2018), deep symbolic regression (Petersen et al., 2021; Mundhenk et al., 2021), and Monte Carlo tree search (Lu et al., 2021; Sun et al., 2023). Sparse regression techniques such as SINDy (Brunton et al., 2016; Rudy et al., 2017; Rao et al., 2023) leverage a predefined library that includes a limited number of candidate terms, which search for the underlying equations in a compact solution space.

**Physics-informed learning.** Physics-informed learning has been developed to deal with noisy and sparse data in the context of equation discovery. Specifically, differentiable models (e.g., neural networks (Raissi et al., 2019; Chen et al., 2021b), cubic splines (Sun et al., 2021; 2022)) are employed to reconstruct the system states and approximate the required derivative terms required to form the underlying eauqtions.

**Vision-based discovery of dynamics.** Very recently, attempts have been made to discover the governing of equations for moving objects directly from videos. These methods are generally based

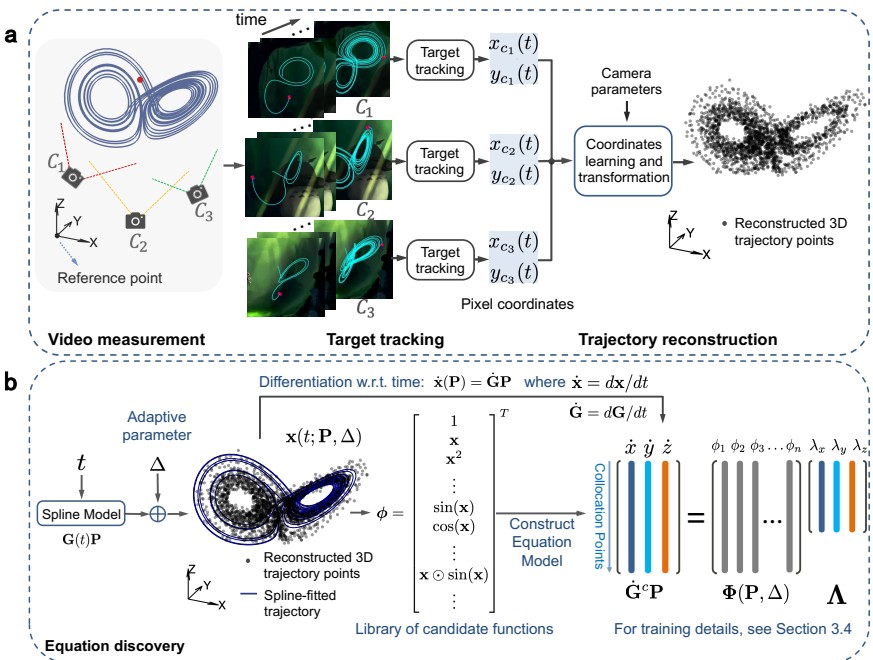

Figure 1: Schematic of vision-based discovery of nonlinear dynamics for 3D moving target. Firstly, we record the motion trajectory of the object in a 3D space using multiple cameras in a predefined reference coordinate system (see **a**). Pixel trajectory coordinates are obtained through target identification and tracking. Note that camera parameters include the camera's position, the normal vector of the camera's view plane, and the calibrated camera parameters, which comprise the scaling factor and tilt angle. In particular, we use coordinate learning and transformation to obtain the spatial motion trajectory in the reference coordinate system. Secondly, for each dimension of the trajectory, we introduce a spline-enhanced library-based sparse regressor to uncover the underlying governing law of dynamics (see **b**).

on autoencoders that extract the latent dynamics for equation discovery (Champion et al., 2019; Udrescu & Tegmark, 2021; Luan et al., 2022). Other related works include the discovery of intrinsic dynamics Floryan & Graham (2022) or fundamental variables (Chen et al., 2022) based on high-dimensional data such as videos.

## 3 METHODOLOGY

We here elucidate the concept and approach of vision-based discovery of nonlinear dynamics for 3D moving target. Figure 1 shows the schematic architecture of our method. The target tracking module serves as the foundational stage, which extracts pixel-level motion information from the target's movements across consecutive frames in a video sequence. The coordinate transformation module utilizes Rodrigues' rotation formula with respect to a predefined reference coordinate origin, which lays the groundwork for the subsequent analysis of the object's dynamics. The final crucial component is the spline-enhanced library-based sparse regressor, essential for revealing the fundamental dynamics governing object motion. We then introduce each module in detail as follows.

### 3.1 COORDINATES TRANSFORMATION

In this paper, we employ three fixed-position cameras oriented in different directions to independently capture the motion of an object (see Figure 1a). With the constraint of calibrating only one camera, our coordinate learning module becomes essential (e.g., the 3D trajectory of the target and other camera parameters can be simultaneously learned). In particular, it is tasked with learning the unknown parameters of the other two cameras, including scaling factors and the rotation angle on each camera plane. These parameters enable to reconstruct the motion trajectory of the object in the reference coordinate system. We leverage Rodrigues' rotation formula to compute vector rotations in three-dimensional space, which enables the derivation of the rotation matrix, describing the rotation operation from a given initial vector to a desired target vector. This formula finds extensive utility in computer graphics, computer vision, robotics, and 3D rigid body motion problems.

In a 3D space, a rotation matrix is used to represent the transformation of a rigid body around an axis. Let $\mathbf{v}_0$ represent the initial vector and $\mathbf{v}_1$ denote the target vector. We denote the rotation matrix as $\mathbf{R}$. The relationship between the pre- and post-rotation vectors can be expressed as $\mathbf{v}_1 = \mathbf{R}\mathbf{v}_0$. The rotation angle, denoted as $\theta$, can be calculated via $\cos\theta = \frac{\mathbf{v}_0 \cdot \mathbf{v}_1}{\|\mathbf{v}_0\|\|\mathbf{v}_1\|}$. The rotation axis is represented by the unit vector $\mathbf{u} = [u_x, u_y, u_z]$, namely, $\mathbf{u} = \frac{\mathbf{v_0} \times \mathbf{v_1}}{\|\mathbf{v_0} \times \mathbf{v_1}\|}$. Having defined the rotation angle $\theta$ and the unit vector $\mathbf{k}$, we construct the rotation matrix $\mathbf{R}$ using Rodrigues' rotation formula, given by

$$\mathbf{R} = \mathbf{I} + \sin\theta \mathbf{U} + (1 - \cos\theta)\mathbf{U}^2. \tag{1}$$

where $\mathbf{I}$ is a $3 \times 3$ identity matrix, and $\mathbf{U}$ is a $3 \times 3$ skew-symmetric matrix representing the cross product of the rotation axis vector $\mathbf{u}$ expressed as

$$\mathbf{U} = \begin{bmatrix} 0 & -u_z & u_y \\ u_z & 0 & -u_x \\ -u_y & u_x & 0 \end{bmatrix}. \tag{2}$$

The resulting expansion of the rotation matrix $\mathbf{R}$ is then given as follows:

$$\mathbf{R} = \begin{bmatrix} \cos\theta + u_x^2(1 - \cos\theta) & u_x u_y(1 - \cos\theta) - u_z \sin\theta & u_x u_z(1 - \cos\theta) + u_y \sin\theta \\ u_x u_y(1 - \cos\theta) + u_z \sin\theta & \cos\theta + u_y^2(1 - \cos\theta) & u_y u_z(1 - \cos\theta) - u_x \sin\theta \\ u_x u_z(1 - \cos\theta) - u_y \sin\theta & u_y u_z(1 - \cos\theta) + u_x \sin\theta & \cos\theta + u_z^2(1 - \cos\theta) \end{bmatrix}. \tag{3}$$

In a predefined reference coordinate system, we use multiple cameras to track the target motion in a 3D space. Let $\mathbf{v}_0 = (0, 0, 1)^T$ be the initial vector, and a camera's plane equation be $Ax + By + Cz + D = 0$, where $\mathbf{v}_1 = (A, B, C)$ is the target vector. The rotation axis vector is $\mathbf{u} = (u_x, u_y, 0)$, and the rotation matrix $R$ is computed as:

$$\mathbf{R} = \begin{bmatrix} \cos\theta + u_x^2(1 - \cos\theta) & u_x u_y(1 - \cos\theta) & u_y \sin\theta \\ u_x u_y(1 - \cos\theta) & \cos\theta + u_y^2(1 - \cos\theta) & -u_x \sin\theta \\ -u_y \sin\theta & u_x \sin\theta & \cos\theta \end{bmatrix}. \tag{4}$$

Let $\mathbf{x}(t) = (x, y, z)^T$ represent the initial coordinates of a moving object at time $t$, and $\mathbf{x}_r = (x_r, y_r, z_r)^T$ denotes its coordinates after rotation. This relationship is defined as $\mathbf{R} \cdot \mathbf{x} = \mathbf{x}_r$, where the rotation matrix $\mathbf{R}$ transforms the initial coordinates $\mathbf{x}$ into the rotated coordinates $\mathbf{x}_r$. Hence, we can obtain $[x_r, y_r, z_r]^\mathrm{T} = \mathbf{R} \cdot [x, y, z]^\mathrm{T}$.

The camera plane equation is $Ax + By + Cz + D = 0$. When projecting a three-dimensional object onto this plane, denoting the coordinates of the projection as $\mathbf{x}_p = (x_p, y_p, z_p)^T$, we have:

$$\mathbf{x}_p = \frac{1}{A^2 + B^2 + C^2} \cdot \begin{bmatrix} x(B^2 + C^2) - A(By + Cz + D) \\ y(A^2 + C^2) - B(Ax + Cz + D) \\ z(A^2 + B^2) - C(Ax + By + D) \end{bmatrix}. \tag{5}$$

If the camera plane's normal vector remains constant and scaling is neglected, the captured trajectory is determined by the camera plane's normal vector, irrespective of its position. After rotating the camera's normal vector $\mathbf{v}_1$ to $\mathbf{v}_0 = (0, 0, 1)^T$ (initial vector), denoted as $\mathbf{x}_{rp} = (x_{rp}, y_{rp}, z_{rp})^T$, we have $\mathbf{x}_{rp} = \mathbf{R} \cdot \mathbf{x}_p$. Here, $x_{rp}$ and $y_{rp}$ depending solely on the camera plane's normal vector $\mathbf{v}_1$ and independent of the parameter $D$ shown in the camera's plane equation. For more details, please refer to Appendix A. In the reference coordinate system, an object's trajectory is projected onto a camera plane. Considering a camera position as the origin of the 2D camera plane, the positional offset of an object's projection on the 2D camera plane with respect to this origin is denoted as $\Delta_c = (\delta_x, \delta_y)^T$. This offset distance $\Delta_c$ can be computed following the details shown in Appendix B.

### 3.2 CUBIC B-SPLINES

B-splines are differentiable, and constructed using piecewise polynomial functions called basis functions. Sun et al. (2021) demonstrated that, when the measurement data is noisy and sparse, cubic B-splines could serve as a differentiable surrogate model to form robust physics-informed learning for equation discovery. We herein adopt this approach to tackle challenges associated with data noise and gaps induced by the imprecise target tracking for discovering laws of 3D dynamics. The $i$-th cubic B-spline basis function of degree $k$, written as $G_{i,k}(u)$, can be defined recursively as:

$$\begin{aligned} G_{i,0}(u) &= \begin{cases} 1 & \text{if } u_i \leq u < u_{i+1} \\ 0 & \text{otherwise} \end{cases}, \\ G_{i,k}(u) &= \frac{u - u_i}{u_{i+k} - u_i} G_{i,k-1}(u) + \frac{u_{i+k+1} - u}{u_{i+k+1} - u_{i+1}} G_{i+1,k-1}(u), \end{aligned} \tag{6}$$

where $u_i$ represents a knot that partitions the computational domain. By selecting appropriate control points and combinations of basis functions, cubic B-splines with $\mathbb{C}^2$ continuity can be customized to meet specific requirements. In general, a cubic B-spline curve of degree $p$ defined by $n + 1$ control points $\mathbf{P} = \{\mathbf{p}_0, \mathbf{p}_1, ..., \mathbf{p}_n\}$ and a knot vector $U = \{u_0, u_1, ..., u_m\}$ is given by: $C(u) = \sum_{i=0}^{n} G_{i,3}(u) \cdot \mathbf{p}_i$. To ensure the curve possesses continuous and smooth tangent directions at the starting and ending nodes, meeting the first derivative interpolation requirement, we use Clamped cubic B-spline curves for fitting.

### 3.3 NETWORK ARCHITECTURE

We utilized the YOLO-v8 for object tracking in the recorded videos (see Figure 1a). Regardless of whether the captured object has an irregular shape or is in a rotated state, we only need to capture their centroid positions and track them to obtain pixel data. Subsequently, leveraging Rodrigues' rotation formula and based on the calibrated camera, we derive the scaling and rotation factors of the other two cameras. These factors enable the conversion of the object trajectory's pixel coordinates into the world coordinates deducing the physical trajectory. For the trajectory varying with time in each dimension, we use the cubic B-splines to fit the trajectory and a library-based sparse regressor to uncover the underlying governing law of dynamics in the reference coordinate system. This approach is capable of dealing with data noise, multiple instances of data missing and gaps.

**Learning 3D trajectory.** In this work, we use a three-camera setup to capture and represent the object's 2D motion trajectory in each video scene, yielding the 2D coordinates denoted as $(x_{rp}, y_{rp})$. The rotation matrix $\mathbf{R}$ is decomposed to retain only the first two rows, denoted as $\mathbf{R}^-$, to suitably handle the projection onto the image planes. Under the condition of calibrating only one camera, we can reconstruct the coordinates of a moving object in the reference 3D coordinate system using three fixed cameras capturing an object's motion in a 3D space. The assumed given information includes normal vectors $\mathbf{v}_1, \mathbf{v}_2, \mathbf{v}_3$ of camera planes for all three cameras, the positions of the cameras, as well as a scaling factor $s_1$ and rotation angles $\vartheta_1$ for one of the cameras. We define the scaling factor vector as $\mathbf{s} = \{s_1, s_2, s_3\}$ and the rotation angle vector as $\vartheta = \{\vartheta_1, \vartheta_2, \vartheta_3\}$. The loss function for reconstructing the 3D coordinates of the object in the reference coordinate system is given by

$$\mathcal{L}_r\left(\mathbf{s}^*; \vartheta^*\right) = \frac{1}{N_m} \left\| \mathbf{R}_1^- \left[ \begin{array}{c} \mathbf{R}_2^- \\ \mathbf{R}_3^- \end{array} \right]^{-1} \left[ \begin{array}{c} s_2 \mathcal{T}\left(\vartheta_2\right) \mathbf{x}_{c_2} + \Delta_{c_2} \\ s_3 \mathcal{T}\left(\vartheta_3\right) \mathbf{x}_{c_3} + \Delta_{c_3} \end{array} \right] - \left(s_1 \mathcal{T}\left(\vartheta_1\right) \mathbf{x}_{c_1} + \Delta_{c_1}\right) \right\|_2^2, \quad (7)$$

where $\mathbf{s}^* = \{s_2, s_3\}$ and $\vartheta^* = \{\vartheta_2, \vartheta_3\}$. Here, $\mathbf{x}_{c_i} = (x_{c_i}, y_{c_i})^T$ represents the pixel coordinates, $N_m$ the number of effectively recognized object coordinate points, $\mathcal{T}(\vartheta)$ the transformation matrix induced by rotation angle $\vartheta$ expressed as $\mathcal{T}(\vartheta) = [\cos\vartheta \ \sin\vartheta; \ -\sin\vartheta \ \cos\vartheta]$.

In the case of using three cameras, the transformation between the object's coordinates $\mathbf{x}_{ref}$ in the reference coordinate system and the pixel coordinates $\mathbf{x}_c$ in the camera setups is as follows

$$\mathbf{x} = \left[ \begin{array}{c} \mathbf{R}_1^- \\ \mathbf{R}_2^- \\ \mathbf{R}_3^- \end{array} \right]^{-1} \left[ \begin{array}{c} s_1 \mathcal{T}(\vartheta_1)\mathbf{x}_{c_1} + \Delta_{c_1} \\ s_2 \mathcal{T}(\vartheta_2)\mathbf{x}_{c_2} + \Delta_{c_2} \\ s_3 \mathcal{T}(\vartheta_3)\mathbf{x}_{c_3} + \Delta_{c_3} \end{array} \right]. \quad (8)$$

By solving for parameter values $(\mathbf{s}^*, \vartheta^*)$ via optimization of Eq. (7), we can subsequently compute the reconstructed 3D physical coordinates through the calculation provided in Eq. (8).

**Equation discovery.** Given the potential challenges in target tracking, e.g., momentary target loss, noise, or occlusions, we leverage physics-informed spline learning to address these issues (see Figure 1b). In particular, cubic B-splines are employed to approximate the 3D trajectory. Given three sets of control points denoted as $\mathbf{P} = \{\mathbf{p}_1, \mathbf{p}_2, \mathbf{p}_3\} \in \mathbb{R}^{r \times 3}$. Given that the coordinate system is arbitrarily defined, and to enhance the fitting of data $\mathcal{D}_r$, we introduce the learnable adaptive offset parameter $\Delta = \{\Delta_1, \Delta_2, \Delta_3\}$. The 3D parametric curves where $\mathbf{x}(t; \mathbf{P}, \Delta)$ are defined by the control point vectors $\mathbf{P}$, the cubic B-spline basis functions $\mathbf{G}(t)$ and the offset parameter $\Delta$, namely, $\mathbf{x}(t; \mathbf{P}, \Delta) = \mathbf{G}(t)\mathbf{P} + \Delta$. Since the basis functions consist of differentiable polynomials, the expression of its differential equation is given by $\dot{\mathbf{x}}(\mathbf{P}) = \dot{\mathbf{G}}\mathbf{P}$. Generally, the dynamics is governed by a limited number of significant terms, which can be selected from a library of $l$ candidate functions, $v.i.z.$, $\phi(\mathbf{x}) \in \mathbb{R}^{1 \times l}$ (Brunton et al., 2016). The governing equations can be written as:

$$\dot{\mathbf{x}}(\mathbf{P}) = \phi(\mathbf{P}, \Delta)\Lambda, \quad (9)$$

where $\phi(\mathbf{P}, \Delta) = \phi(\mathbf{x}(t; \mathbf{P}, \Delta))$, and $\mathbf{\Lambda} = \{\boldsymbol{\lambda}_1, \boldsymbol{\lambda}_2, \boldsymbol{\lambda}_3\} \in \mathcal{S} \subset \mathbb{R}^{l \times 3}$ is the sparse coefficient matrix belonging to a constraint subset $\mathcal{S}$ (only the terms active in $\phi$ exhibit non-zero values).

Accordingly, the task of equation discovery can be formulated as follows: when provided with reconstructed 3D trajectory data $\mathcal{D}_r = \{\mathbf{x}_1^m, \mathbf{x}_2^m, \mathbf{x}_3^m\} \in \mathbb{R}^{N_m \times 3}$. In other words, $\mathcal{D}_r$ is presented as effectively tracking the object movements in a video and subsequently transforming them into a 3D trajectory, where $N_m$ is the number of data points. Our goal is to identify the suitable set of $\mathbf{P}$ and a sparse $\mathbf{\Lambda}$ that fits the trajectory data meanwhile satisfying Eq. (9). Considering that the reconstructed trajectory $\mathcal{D}_r$ might exhibit noise or temporal discontinuity, we use collocation points denoted as $\mathcal{D}_c = \{t_0, t_1, \ldots, t_{n_c-1}\}$ to compensate data imperfection, where $\mathcal{D}_c$ denotes the randomly sampled set of $N_c$ number of collocation points ($N_c \gg N_m$). These points are strategically employed to reinforce the fulfillment of physics constraints at all time instances (see Figure 1b).

### 3.4 NETWORK TRAINING

The loss function for this network comprises three main components, namely, the data component $\mathcal{L}_d$, the physics component $\mathcal{L}_p$, and the sparsity regularizer, given by:

$$\mathcal{L}(\mathbf{P}^*, \mathbf{\Lambda}^*, \Delta^*) = \arg\min_{\{\mathbf{P}, \mathbf{\Lambda}, \Delta\}} [\mathcal{L}_d(\mathbf{P}, \Delta; \mathcal{D}_r) + \alpha \mathcal{L}_p(\mathbf{P}, \Delta, \mathbf{\Lambda}; \mathcal{D}_c)] + \beta \|\mathbf{\Lambda}\|_0, \quad (10)$$

$$\mathcal{L}_d = \frac{1}{N_m} \sum_{i=1}^{3} \|\mathbf{G}_m \mathbf{p}_i + \Delta_i - \mathbf{x}_i^m\|_2^2, \quad \mathcal{L}_p = \frac{1}{N_c} \sum_{i=1}^{3} \left\|\mathbf{\Phi}(\mathbf{P}, \Delta) \boldsymbol{\lambda}_i - \dot{\mathbf{G}}^c \mathbf{p}_i\right\|_2^2. \quad (11)$$

where $\mathbf{G}_m$ denotes the spline basis matrix evaluated at the measured time instances, $\mathbf{x}_i^m$ the coordinates in each dimension after 3D reconstruction in the reference coordinate system (may be sparse or exhibit data gaps whereas $\dot{\mathbf{G}}_c$ the derivative of the spline basis matrix evaluated at the collocation instances. The term $\mathbf{G}_m \mathbf{p}_i$ is employed to fit the measured trajectory in each dimension, while $\dot{\mathbf{G}}^c \mathbf{p}_i$ is used to reconstruct the potential equations evaluated at the collocation instances. Additionally, $\mathbf{\Phi} \in \mathbb{R}^{N_c \times l}$ represents the collocation library matrix encompassing the collection of candidate terms, $\|\mathbf{\Lambda}\|_0$ the sparsity regularizer, $\alpha$ and $\beta$ the relative weighting parameters.

Since the regularizer $\|\mathbf{\Lambda}\|_0$ leads to an NP-hard optimization issue, we apply an Alternate Direction Optimization (ADO) strategy (see Appendix C) to optimize the loss function (Chen et al., 2021b; Sun et al., 2021). The interplay of spline interpolation and sparse equations yields subsequent effects: the spline interpolation ensures accurate modeling of the system's response, its derivatives, and the candidate function terms, thereby laying the foundation for constructing the governing equations. Simultaneously, the equations represented in a sparse manner synergistically constrain spline interpolation and facilitate the projection of accurate candidate functions. Ultimately, this transforms the extraction of a 3D trajectory of an object from video into a closed-form differential equation.

$$\dot{\mathbf{x}} = \phi(\mathbf{x} - \Delta^*) \mathbf{\Lambda}^*. \quad (12)$$

After applying ADO to execute our model, resulting in the optimal control point matrix $\mathbf{P}^*$, sparse matrix $\mathbf{\Lambda}^*$, and adaptive parameter $\Delta^*$, an affine transformation is necessary to eliminate $\Delta^*$ in the identified equations. We replace $\mathbf{x}$ with $\mathbf{x} - \Delta^*$, as shown in Eq. (12), to obtain the final form of equations. We then assign a small value to prune equation coefficients, yielding the discovered governing equations in a predefined 3D coordinate system.

## 4 EXPERIMENTS

In this section, we evaluate our method for uncovering 3D governing equations of a moving target automatically from videos using nine datasets[1]. The nonlinear dynamical equations for these chaotic systems and their respective trajectories can be found in Appendix D (see Figure S1). We generate 3D trajectories based on the governing equations of the dataset and subsequently produce corresponding video data captured from various positions. Our analysis encompasses the method's robustness across distinct video backgrounds, varying shapes of moving objects, object rotations, levels of data noise, and occlusion scenarios. We further validate the identified equations demonstrating their interpretability and generalizability. The proposed computational framework is implemented in PyTorch. All simulations in this study are conducted on an Intel Core i9-13900 CPU workstation with an NVIDIA GeForce RTX 4090 GPU.

---

[1]The datasets are derived from instances introduced in Gilpin (2021), where we utilize the following examples: Lorenz, SprottE, RayleighBenard, SprottF, NoseHoover, Tsucs2 and WangSun.

**Data generation.** The videos in this study are synthetically generated using MATLAB to simulate real dynamic systems captured by cameras. To commence, the dynamic system is pre-defined, and its trajectory is simulated utilizing the 4th-order Runge-Kutta method in MATLAB. Leveraging the generated 3D trajectory, a camera's orientation is established within a manually defined 3D coordinate system to simulate the 2D projection of the object onto the camera plane. The original colored images featuring the moving object are confined to dimensions of $512 \times 512$ pixels at 25 frames per second (fps). Various shapes are employed as target markers in the video along with local dynamics (e.g., with self-rotation) to emulate the motion of the object (see Appendix E). Subsequently, a set of background images are randomly selected to mimic the real-world video scenarios. The resultant videos generated within the background imagery comprise color content, with each frame containing RGB channels (e.g., see Appendix Figure S2). After obtaining the video data, it becomes imperative to perform object recognition and tracking on the observed entities based on the YOLO-v8 method.

## 4.1 RESULTS

**Evaluation metrics.** We employ both qualitative and quantitative metrics to assess the performance of our method. Our goal is to identify all equation terms as accurately as possible while eliminating irrelevant terms (False Positives) to the greatest extent. The error $\ell_2$, represented as $||\mathbf{\Lambda}_{id} - \mathbf{\Lambda}_{true}||_2 / ||\mathbf{\Lambda}_{true}||_2$, quantifies the relative difference between the identified coefficients $\mathbf{\Lambda}_{id}$ and the ground truth $\mathbf{\Lambda}_{true}$. To avoid the overshadowing of smaller coefficients when there is a significant disparity in their magnitudes, we introduce a non-dimensional measure to obtain a more comprehensive evaluation.

Table 1: The performance of our method compared to the PySINDy in reconstructing three-dimensional coordinates from videos.

| Cases | Methods | Terms Found? | False Positives | $\ell_2$ Error ($\times 10^{-2}$) | $P$ (%) | $R$ (%) |
|---|---|---|---|---|---|---|
| Lorenz | Ours | Yes | 1 | **1.50** | **92.31** | **100** |
| | PySINDy | Yes | 1 | 4.17 | **92.31** | **100** |
| SprottE | Ours | Yes | **0** | **0.15** | **100** | **100** |
| | PySINDy | Yes | 3 | 3.48 | 72.73 | **100** |
| RayleighBenard | Ours | Yes | 1 | **2.00** | **91.67** | **100** |
| | PySINDy | Yes | 2 | 2.74 | 84.62 | **100** |
| SprottF | Ours | **Yes** | **0** | **0.16** | **100** | **100** |
| | PySINDy | No | 1 | 7.51 | 90 | 90 |
| NoseHoover | Ours | Yes | **0** | **6.22** | **100** | **100** |
| | PySINDy | Yes | 3 | 824.44 | 75 | **100** |
| Tsucs2 | Ours | Yes | 1 | **5.39** | **93.75** | **100** |
| | PySINDy | Yes | 1 | 12.29 | **93.75** | **100** |
| WangSun | Ours | **Yes** | 1 | **0.16** | **93.33** | **100** |
| | PySINDy | No | 3 | 856.47 | 86.67 | 92.86 |

The discovery of governing equations can be framed as a binary classification task (Rao et al., 2022), determining whether a particular term exists or not, given a candidate library. Hence, we introduce precision and recall as metrics for evaluation, which quantify the proportion of correctly identified coefficients among the actual coefficients, expressed as: $P = ||\mathbf{\Lambda}_{\text{id}} \odot \mathbf{\Lambda}_{\text{true}}||_0 / ||\mathbf{\Lambda}_{\text{id}}||_0$ and $R = ||\mathbf{\Lambda}_{\text{id}} \odot \mathbf{\Lambda}_{\text{true}}||_0 / ||\mathbf{\Lambda}_{\text{true}}||_0$, where $\odot$ denotes element-wise product. Successful identification is achieved when both the entries in the identified and true vectors are non-zero.

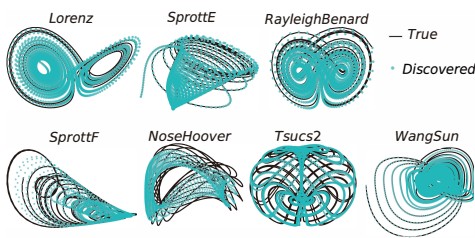

Figure 2: Comparison between the discovered 3D trajectories and the ground truth.

**Discovery results.** Based on our evaluation metrics (e.g., the $\ell_2$ error, the number of correct and incorrect equations terms found, precision, and recall), a detailed analysis of the experimental results obtained by our method is found in Table 1 (without data noise). After reconstructing the 3D trajectories in the world coordinate system, we also compare our approach with PySINDy (Brunton et al., 2016) as the baseline model. The library of candidate functions includes combinations of system states with polynomials up to the third order. The listed results are averaged over five trials. It demonstrates that our method outperforms PySINDy on each dataset in the pre-defined coordinate system. The explicit forms of the discovered governing equations for 3D moving objects obtained using our approach can be further found in Appendix F (e.g., Table S2). It is evident from Appendix Table S2 that the discovered equations by our method align better with the ground truth. We also reconstructed the motion trajectories in a 3D space using our discovered equations compared with the actual trajectories under the same coordinate system, as shown in Figure 2. These two trajectories nearly coincide, demonstrating the feasibility of our method.

It is noted that we also tested the variations and rotations of the moving object shapes in the recorded videos (e.g., see Appendix Figure S2) and found that they have little impact on the performance of

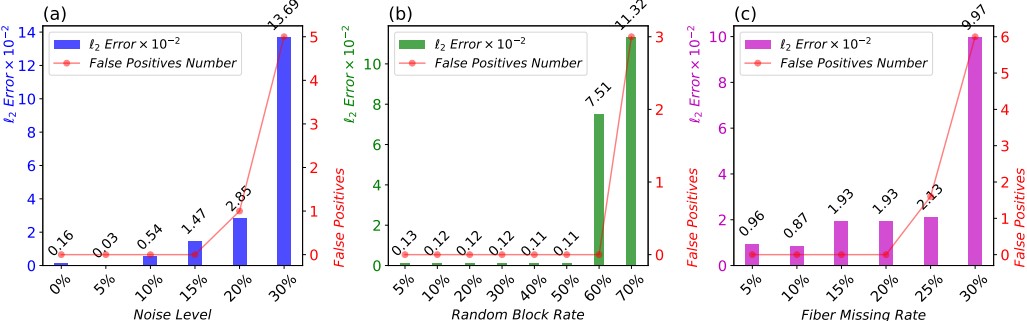

Figure 3: The influence of noisy and missing data (e.g., random block missing and fiber missing) on the experimental results, using the sprootF-based video data as an example. The evaluation metrics include the $l_2$ relative error and the number of incorrectly identified equation coefficients. We analyzed the effect of (a) noise levels, (b) random block missing rates, and (c) fiber missing rates, respectively, to test the model's robustness.

our algorithm, primarily affecting the tracking efficiency. In fact, encountering noise and situations where moving objects are occluded during the measurement process can significantly impact our experimental results. To assess the robustness of our algorithm, we selected the SprottF instance for in-depth analysis and conducted experiments under various noise levels and different data occlusion scenarios. The experimental results are detailed in Figure 3. It is seen that our approach is robust against data noise and missing, discussed in detail as follows.

**Noise effect.** The Gaussian noise with zero mean and unit variance at a given level (e.g., 0%, 5%, ..., 30%) is added to the generated video data. To address the issue of small coefficients being over-shadowed due to significant magnitude differences, we use two evaluation metrics in a standardized coordinate system: the $\ell_2$ error and the count of incorrectly identified equation coefficients. Figure 3a showcases our method's performance across various noise levels. We observe that up to a 20% noise interference, our method almost accurately identifies all correct coefficients of the governing equation. However, beyond a 30% noise level, our method's performance begins to decline.

**Random block missing data effect.** To evaluate our algorithm's robustness in the presence of missing data, we consider two missing scenarios (e.g., the target is blocked in the video scene), namely, random block missing and fiber missing (see Appendix Figure S3 for example). Firstly, we randomly introduce non-overlapping occlusion blocking on the target in the video during the observation period. Each block covers 1% of the total time periods. We validate our method's performance as the number of occlusion blocks increases. The "random block rate" represents the overall occlusion time as a percentage of the total observation time. We showcase our algorithm's robustness by introducing occlusion blocks that temporarily obscure the moving object, rendering it unidentifiable (see Figure 3b). These non-overlapping occlusion blocks progressively increase in number, simulating higher occlusion rates. Remarkably, our algorithm remains highly robust even with up to 50% data loss due to occlusion.

**Fiber missing data effect.** Additionally, we conducted tests for scenarios involving continuous missing data (defined as fiber missing). By introducing 5 non-overlapping occlusion blocks randomly throughout the observation period, we varied the occlusion duration of each block, quantified by the "fiber missing rate" — the ratio of continuous missing data to the overall data volume. In Figure 3c, we explore the impact of increasing occlusion duration per block while maintaining a constant number of randomly selected occlusion blocks. All results are averaged over five trials. Our algorithm demonstrates strong stability even when the fiber missing rate is around 20%.

**Simulating real-world scenario.** Furthermore, we generated a synthetic video dataset simulating real-world scenarios. Here, we modeled the observed object as an irregular shape undergoing random self-rotational motion and size variations, as shown in Appendix Figure S4a. Note that the size variations simulate changes in the camera's focal length when capturing the moving object in depth. The video frames were perturbed with a zero mean Gaussian noise (variance = 0.01). Moreover, a tree-like obstruction was introduced to further simulate the real-world complexity (e.g., the object might be obscured during motion) as depicted in Appendix Figure S4b. Despite these challenges, our method can discover the governing equations of the moving object in the reference coordinate system, showing its potential in practical applications. Please refer to Appendix G for more details.

Overall, our algorithm proves robust in scenarios with unexpected data noise, multiple instances of data loss, and continuous data gaps, effectively uncovering the underlying governing laws of dynamics for the moving object in a 3D space based on raw videos.

## 4.2 ABLATION STUDY

We performed an ablation study to validate whether the physics component in the spline-enhanced library-based sparse regressor module is effective. Hence, we introduced an ablated model named Model-A (e.g., fully decoupled "spline + SINDy" approach). We first employed the cubic splines to interpolate the 3D trajectory in each dimension and then computed the time derivatives of the fitted trajectory points based on

Table 2: Test results for the ablated model named Model-A (i.e., spline + SINDy) under varying noise levels, random block rates, and fiber missing rates on discovering the SprottF equations.

| Conditions | Rate (%) | Methods | Terms Found? | False Positives | $\ell_2$ Error $(\times 10^{-2})$ | $P$ (%) | $R$ (%) |
|---|---|---|---|---|---|---|---|
| Noise | 10 | Ours | **Yes** | **0** | **0.77** | **100** | **100** |
| | | Model-A | No | 1 | 8.78 | 90 | 90 |
| | 20 | Ours | **Yes** | 1 | **2.85** | **100** | **100** |
| | | Model-A | No | 1 | 17.79 | 80 | 80 |
| Random Block | 10 | Ours | **Yes** | **0** | **0.77** | **100** | **100** |
| | | Model-A | No | 3 | 9.49 | 75 | 90 |
| | 20 | Ours | **Yes** | **0** | **2.19** | **100** | **100** |
| | | Model-A | No | 1 | 7.67 | 90 | 90 |
| Fiber Missing | 10 | Ours | **Yes** | **0** | **0.87** | **100** | **100** |
| | | Model-A | No | 3 | 7.88 | 75 | 90 |
| | 20 | Ours | **Yes** | **0** | **1.71** | **100** | **100** |
| | | Model-A | No | 4 | 10.79 | 66.67 | 80 |

spline differentiation. These trajectories and the estimated derivatives are then fed into the SINDy model for equation discovery. Taking the instance of SprootF as an example, we show in Table 2 the performance of the ablated model under varying noise levels, random block rates, and fiber missing rates. It is observed that the performance of the ablated model deteriorates in all considered cases. Hence, we can ascertain that the physics-informed spline learning in the library-based sparse regressor module plays a crucial role in equation discovery under imperfect data conditions.

## 4.3 DISCUSSION AND LIMITATIONS

The above results show that our approach can effectively uncover the governing equations of a moving target in a 3D space directly from a set of recorded videos. The false positives of identification, when in the presence (e.g., see Appendix Table S2), are all small constants. We consider these errors to be within a reasonable range. This is because the camera pixels can only take approximate integer values, and factors such as the size of pixels captured by the camera and the number of cameras capturing the moving object can affect the reconstruction of the 3D coordinates in the reference coordinate system. The experimental results can be further improved when high-resolution videos are recorded and more cameras are used. There is an affine transformation relationship between the artificially set reference coordinate system and the actual coordinate system. Potential errors in learning such a relationship also lead to false positives in governing equation discovery.

Despite efficacy, our approach has limitations in certain scenarios. For instance, the library-based sparse regression technique encounters a bottleneck when identifying very complex equations when the a priori knowledge of the candidate terms is deficient. We plan to integrate symbolic regression techniques to tackle this challenge. Furthermore, the present study only focuses on discovering the 3D dynamics of a single moving target in a video scene. In the future, we will test the discovery of dynamics for multiple moving objects (inter-coupled or independent).

## 5 CONCLUSION

We proposed a vision-based method to distill the governing equations for nonlinear dynamics of a moving object in a 3D space, solely from video data captured by a set of three cameras. By leveraging geometric transformations in a 3D space, combined with Rodrigues' rotation formula and computer vision techniques to track the object's motion, we can learn and reconstruct the 3D coordinates of the moving object in a user-defined coordinate system with the calibration of only one camera. Building upon this, we introduced an adaptive spline learning framework integrated with a library-based sparse regressor to identify the underlying law of dynamics. This framework can effectively handle challenges posed by partially missing and noisy data, successfully uncovering the governing equations of the moving target in a predefined reference coordinate system. The efficacy of this method has been validated on synthetic videos that record the behavior of different nonlinear dynamic systems. This approach offers a novel perspective for understanding the complex dynamics of moving objects in a 3D space. We will test it on real-world recorded videos in our future study.

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

APPENDIX

This appendix provides extrat explanations for several critical aspects, including method validation and computation, data generation, and object recognition and tracking.

## A  PLANE PROJECTION PROOF

In a 3D space, let us assume a camera plane equation is represented as $Ax + By + Cz + D = 0$. We then expand the equation $\mathbf{x}_{rp} = \mathbf{R} \cdot \mathbf{x}_p$ and obtain

$$
\begin{bmatrix} x_{rp} \\ y_{rp} \\ z_{rp} \end{bmatrix} = \begin{bmatrix} \cos\theta + u_x^2(1 - \cos\theta) & u_x u_y(1 - \cos\theta) & u_y \sin\theta \\ u_x u_y(1 - \cos\theta) & \cos\theta + u_y^2(1 - \cos\theta) & u_x \sin\theta \\ -u_y \sin\theta & u_x \sin\theta & \cos\theta \end{bmatrix}
$$
$$
\cdot \begin{bmatrix} x(B^2 + C^2) - A(By + Cz + D) \\ y(A^2 + C^2) - B(Ax + Cz + D) \\ z(A^2 + B^2) - C(Ax + By + D) \end{bmatrix} \cdot \frac{1}{A^2 + B^2 + C^2} \tag{S1}
$$

The expansion of $x_{rp}$ in the above equation reads:

$$
\begin{aligned}
x_{rp} = &[\cos\theta + u_x^2(1 - \cos\theta)] \cdot \frac{x(B^2 + C^2) - A(By + Cz + D)}{A^2 + B^2 + C^2} \\
&+ u_x u_y(1 - \cos\theta) \cdot \frac{y(A^2 + C^2) - B(Ax + Cz + D)}{A^2 + B^2 + C^2} \\
&+ (u_y \sin\theta) \cdot \frac{z(A^2 + B^2) - C(Ax + By + D)}{A^2 + B^2 + C^2}
\end{aligned} \tag{S2}
$$

Collecting the coefficients of $D$ from Eq. (S2) yields zero, namely,

$$
\begin{aligned}
&- A[\cos\theta + u_x^2(1 - \cos\theta)] - Cu_y \sin\theta - Bu_y u_x(1 - \cos\theta) \\
&= \frac{AC}{\sqrt{A^2 + B^2 + C^2}} - \frac{AC}{\sqrt{A^2 + B^2 + C^2}} + \frac{AB^2(\frac{C}{\sqrt{A^2+B^2+C^2}} - 1)}{A^2 + B^2} - \frac{AB^2(\frac{C}{\sqrt{A^2+B^2+C^2}} - 1)}{A^2 + B^2} \\
&= 0
\end{aligned} \tag{S3}
$$

Similarly, it can be proven that the coordinate in $y_{rp}$ is also independent of the parameter $D$.

## B  COMPUTE THE OFFSET DISTANCE OF A CAMERA PLANE

As shown in Appendix A, when the camera plane's normal vector remains unchanged, the shape of the recorded motion trajectory remains constant. However, due to the camera's position not aligning with the origin of the reference coordinate system, there will be positional offsets relative to trajectories recorded by camera planes with the same normal vector but positioned at the coordinate origin. To calculate these offset distances, the approach involves determining the post-rotation coordinate plane and subsequently computing the point-to-plane distance. In this regard, we employ a method based on three-point plane determination: given three points $p_1(x_1, y_1, z_1)$, $p_2(x_2, y_2, z_2)$ and $p_3(x_3, y_3, z_3)$, the primary task is to ascertain the plane's equation, with a key focus on deriving one of the plane's normal vector $\vec{n}$ given by

$$
\vec{n} = p_1 p_2 \times p_1 p_3 = \begin{vmatrix} i & j & k \\ x_2 - x_1 & y_2 - y_1 & z_2 - z_1 \\ x_3 - x_1 & y_3 - y_1 & z_3 - z_1 \end{vmatrix} = ai + bj + ck = (a, b, c)^T. \tag{S4}
$$

A visual representation of this process is shown in Figure S1. The offset relative to the origin on the 2D plane can be obtained by computing the distance of the position of the camera in the 3D space to the $X'OZ'$ plane and its distance to the $X'OY'$ plane.

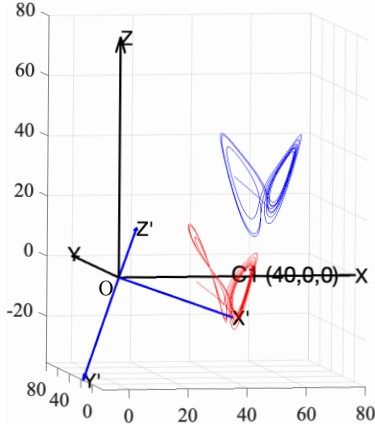

Figure S1: Schematic of trajectory projection from the 3D space to a 2D plane. The blue trajectory represents the 3D motion trajectory, while the red trajectory represents its projection on the 2D camera plane. Here, $C1$ denotes the position of the camera. The normal vector of the camera plane $X'OY'$ is denoted as $Z'$.

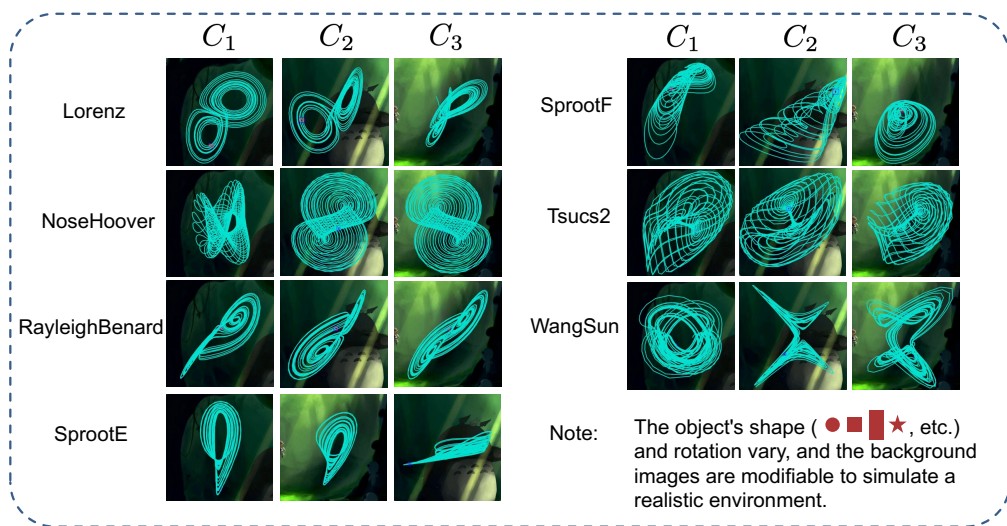

Figure S2: Example of generated video data. In the simulated videos capturing the 3D object motion, we use three cameras, $C_1$, $C_2$, and $C_3$, at different positions and orientations to record the object's trajectory.

## C    ALTERNATE DIRECTION OPTIMIZATION

Addressing the optimization problem given in Eq. (10) directly via gradient descent is highly challenging, given the NP-hard problem induced by the $\ell_0$ regularizer. Alternatively, relaxing $\ell_0$ to $\ell_1$ eases the optimization process but only provides a loose promotion of sparsity. We employ an alternating direction optimization (ADO) strategy that hybridizes gradient descent optimization and sparse regression. The approach involves decomposing the overall optimization problem into a set of tractable sub-optimization problems, formulated as follows:

$$\boldsymbol{\lambda}_i^{(k+1)} := \arg\min_{\boldsymbol{\lambda}_i} \left\| \boldsymbol{\Phi}\left(\mathbf{P}^{(k)}, \Delta^{(k)}\right)\boldsymbol{\lambda}_i - \dot{\mathbf{G}}^c \mathbf{p}_i^{(k)} \right\|_2^2 + \beta \|\boldsymbol{\lambda}_i\|_0, \tag{S5}$$

$$\left\{ \mathbf{P}^{(k+1)}, \tilde{\boldsymbol{\Lambda}}^{(k+1)}, \Delta^{(k+1)} \right\} := \arg\min_{\{\mathbf{P}, \tilde{\boldsymbol{\Lambda}}, \Delta\}} \left[ \mathcal{L}_d(\mathbf{P}, \Delta) + \alpha \mathcal{L}_p(\mathbf{P}, \Delta, \tilde{\boldsymbol{\Lambda}}) \right], \tag{S6}$$

where $k$ denotes the index of the alternating iteration, $\tilde{\boldsymbol{\Lambda}}$ consists of only the non-zero coefficients in $\tilde{\boldsymbol{\Lambda}}^{(k+1)} = \left\{ \boldsymbol{\lambda}_1^{(k+1)}, \boldsymbol{\lambda}_2^{(k+1)}, \boldsymbol{\lambda}_3^{(k+1)} \right\}$. In each iteration, $\boldsymbol{\Lambda}^{(k+1)}$ (shown in Eq. (S5)) is determined

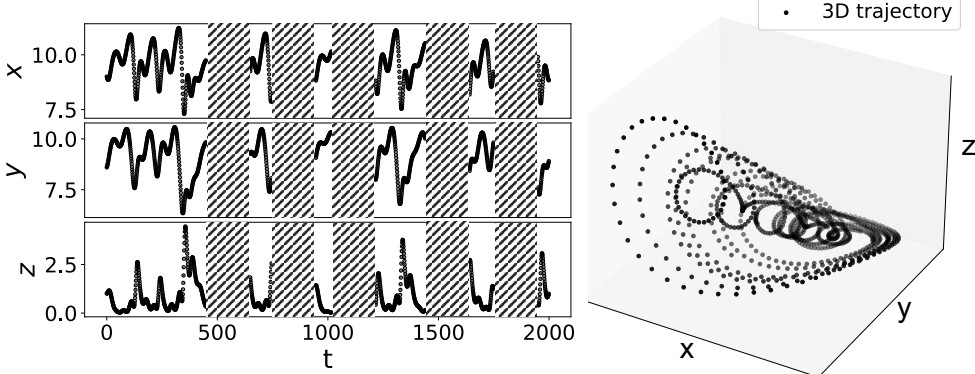

Figure S3: Example of the 3D trajectory reconstruction of the observed object under conditions of occlusion-induced data missing (e.g., fiber missing). The shading areas in the left figure represent regions affected by occlusion.

using the STRidge algorithm (Rudy et al., 2017) with adaptive hard thresholding, pruning small values by assigning zero. The optimization problem in Eq. (S6) can be solved via gradient descent to obtain the updated $\mathbf{P}^{(k+1)}$, $\tilde{\mathbf{\Lambda}}^{(k+1)}$ and $\Delta^{(k+1)}$ with the remaining terms in $\mathbf{\Phi}$ (e.g., redundant terms are pruned). This iterative process continues for multiple iterations until achieving a final balance between the spline interpolation and the pruned equations.

## D CHAOTIC SYSTEMS

In Table S1, we present the original equations for all instances and depict their respective trajectories under given initial conditions and corresponding parameter settings.

## E EXAMPLES OF MEASURED VIDEOS

To simulate real-world scenarios, we considered various scenarios while generating simulated video data. These include different environmental background images, varied shapes of the moving objects, and local self-rotation of the objects. We carefully crafted video data that adheres to these conditions. Figure S2 illustrates the trajectories of the moving objects recorded by the cameras in pixel coordinates. To illustrate scenarios involving the occlusion of moving objects, we employ occlusion blocks to generate missing data. The 3D trajectory data, depicted in Figure S3 for example, corresponds to the reconstruction of an observed object motion in the presence of occlusion-induced data gaps.

## F DISCOVERED EQUATIONS

In Table S2, we compare the discovered equations with the ground truth in the predefined reference coordinate system. The coefficient values are quoted in two decimal places.

## G SIMULATING REALISTIC SCENARIOS

Taking the instance of SprottF as an example, where the initial condition for this experiment is set to be $(-1.17, -1.1, 1)$. As a result, our method effectively reconstructed the 3D trajectory of the moving object, where we see data gaps occur in the time series due to the obstruction. nevertheless, our approach still manages to discover the underlying governing equations even under complex environmental conditions (see Figure S4b).

Table S1: Configuration conditions for various 3D chaotic systems.

| Cases | Original equation | Parameters | Initial condition | True trajectory |
|---|---|---|---|---|
| Lorenz | $\dot{x} = \sigma(y - x)$ 
 $\dot{y} = x(\rho - z) - y$ 
 $\dot{z} = xy - \beta z$ | $\sigma = 10$ 
 $\rho = 28$ 
 $\beta = 8/3$ | $(-8, 7, 27)$ | |
| SprottE | $\dot{x} = yz$ 
 $\dot{y} = x^2 - y$ 
 $\dot{z} = 1 - 4x$ | $n$ | $(-1, 1, 1)$ | |
| RayleighBenard | $\dot{x} = a(y - x)$ 
 $\dot{y} = ry - xz$ 
 $\dot{z} = xy - bz$ | $a = 30$ 
 $b = 5$ 
 $r = 18$ | $(-16, -13, 27)$ | |
| SprottF | $\dot{x} = y + z$ 
 $\dot{y} = -x + ay$ 
 $\dot{z} = x^2 - z$ | $a = 0.5$ | $(-1, -1.4, 1)$ | |
| NoseHoover | $\dot{x} = y$ 
 $\dot{y} = -x + yz$ 
 $\dot{z} = a - y^2$ | $a = 1.5$ | $(-2, 0.5, 1)$ | |
| Tsucs2 | $\dot{x} = a(y - x) + dxz$ 
 $\dot{y} = kx + fy - xz$ 
 $\dot{z} = cz + xy - eps \times x^2$ | $a = 40$ 
 $c = 0.8$ 
 $d = 0.5$ 
 $eps = 0.65$ 
 $f = 20$ 
 $k = 1$ | $(1, 1, 5)$ | |
| WangSun | $\dot{x} = ax + qyz$ 
 $\dot{y} = bx + dy - xz$ 
 $\dot{z} = ez + fxy$ | $a = 0.5$ 
 $b = -1$ 
 $d = -0.5$ 
 $e = -1$ 
 $f = -1$ 
 $q = 1$ | $(1, 1, 0)$ | |

Table S2: Discovered governing equations in the predefined reference coordinate system.

| Cases | Ground truth | Ours | PySINDy |
|---|---|---|---|
| Lorenz | $\dot{x} = -10x + 10y$ 
 $\dot{y} = -xz + 28x - y$ 
 $\quad + 10z - 270$ 
 $\dot{z} = xy - 10x - 10y$ 
 $\quad - 2.667z + 100$ | $\dot{x} = -10.00x + 10.01y - 0.94$ 
 $\dot{y} = -0.99xz + 27.72x - 0.97y$ 
 $\quad + 9.84z - 265.96$ 
 $\dot{z} = 0.99xy - 9.93x - 9.88y$ 
 $\quad - 2.70z + 98.72$ | $\dot{x} = -10.98x + 10.62y + 3.59$ 
 $\dot{y} = -1.05xz + 28.14x - 0.40y$ 
 $\quad + 10.69z - 281.43$ 
 $\dot{z} = 0.98xy - 9.24x - 10.26y$ 
 $\quad - 2.87z + 99.68$ |
| SprottE | $\dot{x} = yz - 10z$ 
 $\dot{y} = x^2 - 20x - y$ 
 $\quad + 110$ 
 $\dot{z} = 41 - 4x$ | $\dot{x} = 1.00yz - 9.95z$ 
 $\dot{y} = 1.00x^2 - 20.13x - 0.93y$ 
 $\quad + 110.09$ 
 $\dot{z} = 41.00 - 4.00x$ | $\dot{x} = 0.98yz - 9.76z$ 
 $\dot{y} = 0.97x^2 - 19.38x - 0.89y$ 
 $\quad - 0.004y^2 + 106.17$ 
 $\dot{z} = 42.45 - 3.98x - 0.31y$ 
 $\quad 0.014y^2$ |
| RayleighBenard | $\dot{x} = -30x + 30y$ 
 $\dot{y} = -xz + 18y + 10z$ 
 $\quad - 180$ 
 $\dot{z} = xy - 10x - 10y$ 
 $\quad - 5z + 100$ | $\dot{x} = -30.07x + 29.45y + 2.22$ 
 $\dot{y} = -1.04xz + 18.14y + 10.28z$ 
 $\quad - 181.00$ 
 $\dot{z} = 0.99xy - 10.13x - 9.25y$ 
 $\quad - 5.04z + 96.69$ | $\dot{x} = -29.82x + 29.22y + 2.15$ 
 $\dot{y} = -1.02xz + 18.14y + 10.09z$ 
 $\quad - 0.39x - 176.69$ 
 $\dot{z} = 0.99xy - 10.36x - 8.82y$ 
 $\quad - 5.04z + 96.04$ |
| SprootF | $\dot{x} = y + z - 10$ 
 $\dot{y} = -x + 0.5y + 5.0$ 
 $\dot{z} = x^2 - 20x - z$ 
 $\quad + 100$ | $\dot{x} = 1.00y + 1.00z - 9.99$ 
 $\dot{y} = -1.00x + 0.50y + 5.01$ 
 $\dot{z} = 1.00x^2 - 20.01x - 1.00z$ 
 $\quad + 100.12$ | $\dot{x} = 0.78y + 0.50x - 2.38$ 
 $\dot{y} = -1.00x + 0.50y + 5.00$ 
 $\dot{z} = 0.99x^2 - 19.89x - 0.99z$ 
 $\quad + 99.49$ |
| NoseHoover | $\dot{x} = y - 10$ 
 $\dot{y} = -x + yz - 10z + 10$ 
 $\dot{z} = -y^2 + 20y - 98.5$ | $\dot{x} = 1.10y - 9.99$ 
 $\dot{y} = -0.91x + 1.07yz - 9.66z$ 
 $\quad + 9.11$ 
 $\dot{z} = -1.14y^2 + 20.66y - 92.27$ | $\dot{x} = 10.97y - 99.45$ 
 $\dot{y} = -9.10x + 10.38yz - 93.92z$ 
 $\quad + 91.85$ 
 $\dot{z} = -11.18y^2 + 202.89y - 0.09x$ 
 $\quad + 0.016x^2 - 0.014xy - 905.85$ |
| Tsucs2 | $\dot{x} = 0.5xz - 40x + 40y$ 
 $\quad - 5z$ 
 $\dot{y} = -1.00xz + 1.00x + 20y$ 
 $\quad + 10z - 210$ 
 $\dot{z} = -0.65x^2 + xy + 3x$ 
 $\quad - 10y + 0.8z + 35$ | $\dot{x} = 0.49xz - 39.58x + 39.92y$ 
 $\quad - 4.94z + 8.45$ 
 $\dot{y} = -0.99xz + 0.63x + 19.95y$ 
 $\quad + 10.01z - 201.61$ 
 $\dot{z} = -0.65x^2 + 0.99xy + 3.48x$ 
 $\quad - 10.258y + 0.76z + 35.19$ | $\dot{x} = 0.46xz - 37.76x + 38.64y$ 
 $\quad - 4.66z + 3.14$ 
 $\dot{y} = -0.92xz - 0.75x + 19.40y$ 
 $\quad + 9.35z - 183.22$ 
 $\dot{z} = -0.59x^2 + 0.90xy + 3.11x$ 
 $\quad - 9.12y + 0.72z - 0.59x^2$ 
 $\quad + 30.91$ |
| WangSun | $\dot{x} = 0.5x + yz - 10 * z$ 
 $\quad - 5.0$ 
 $\dot{y} = -xz - x - 0.5y$ 
 $\quad + 10 * z + 15.0$ 
 $\dot{z} = -xy + 10x + 10y$ 
 $\quad - z - 100$ | $\dot{x} = 0.50x + 1.00yz - 9.97z$ 
 $\quad - 4.89$ 
 $\dot{y} = -1.00xz - 0.99x - 0.50y$ 
 $\quad + 9.99z + 14.90$ 
 $\dot{z} = -1.00xy + 9.98x + 9.99y$ 
 $\quad - 1.00z - 99.94$ | $\dot{x} = 4.89x + 9.77yz - 97.84z$ 
 $\quad - 48.95$ 
 $\dot{y} = -8.19xz - 9.37x + 82.27y$ 
 $\quad 95.25$ 
 $\dot{z} = -9.75xy + 97.65x + 94.74y$ 
 $\quad - 10.02z + 0.15y^2$ 
 $\quad - 0.18z^2 - 963.18$ |

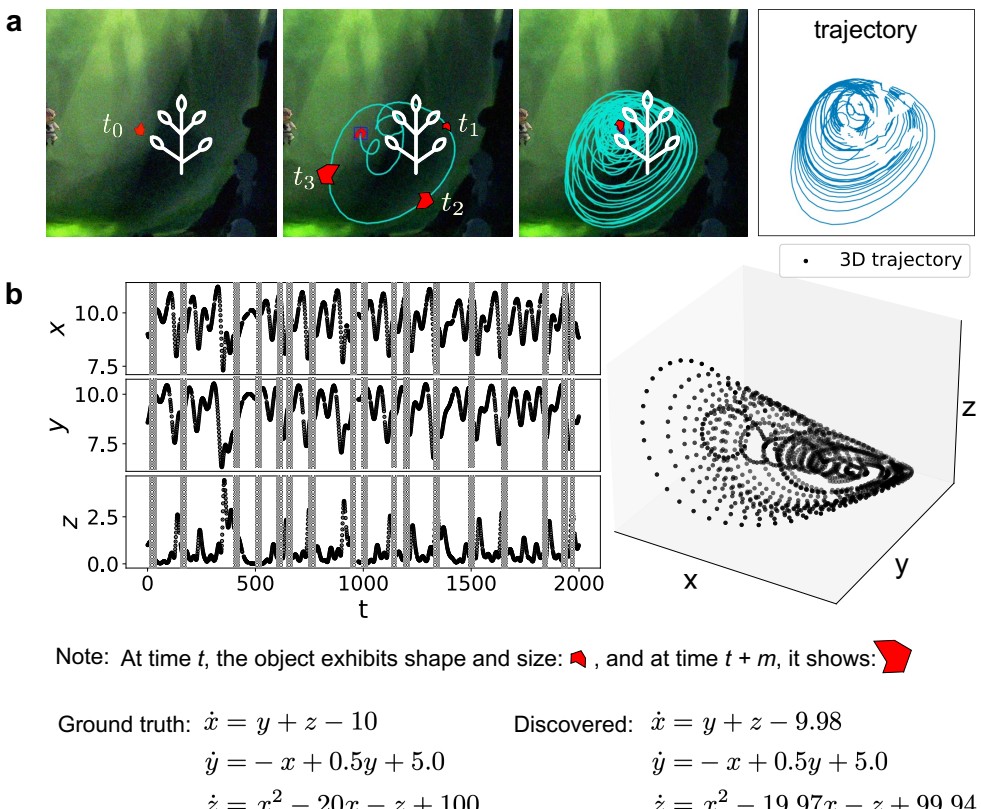

**Note:** At time $t$, the object exhibits shape and size: ◤ , and at time $t + m$, it shows: ⬠

Ground truth:
$$\dot{x} = y + z - 10$$
$$\dot{y} = -x + 0.5y + 5.0$$
$$\dot{z} = x^2 - 20x - z + 100$$

Discovered:
$$\dot{x} = y + z - 9.98$$
$$\dot{y} = -x + 0.5y + 5.0$$
$$\dot{z} = x^2 - 19.97x - z + 99.94$$

Figure S4: Example of a synthetic dataset simulating real-world scenarios. **a**. An example of the generated video for an object with an irregular shape undergoing random self-rotational motion and size variations. The video frames were perturbed with a zero mean Gaussian noise (variance = 0.01), and a tree-like obstruction was introduced to further simulate real-world complexity. **b**. We reconstructed the 3D trajectory of the observed target under conditions of occlusion-induced data missing. The shading areas indicate the regions impacted by the obstruction. Our approach can reconstruct the 3D point trajectories from sparse observation points, revealing accurate discovery of the underlying governing equations. Note that the video file can be found in the supplementary material.

