# OpenReview forum: "Vision-based Discovery of Nonlinear Dynamics for 3D Moving Target"
_ICLR.cc/2024/Conference — ICLR 2024 Conference Withdrawn Submission_

### Official Review · Reviewer_1rkU · 2023-10-31

**Soundness:** 2 fair
**Presentation:** 4 excellent
**Contribution:** 2 fair
**Rating:** 5
**Confidence:** 5

**Summary:**

This paper proposed a method to automatically determine the governing equations of nonlinear dynamics for 3D moving targets via raw videos recorded by a set of cameras.
A set of synthetically generated videos is used to test the performance of the proposed technique.

**Strengths:**

General equations on the 3D motion dynamics are determined from video data of moving objects.

**Weaknesses:**

Only results from synthetic data are provided. Even though some test results with additive noise are presented, they cannot replace experiments with real data, where additional factors such as changes in illumination and localization errors may take place. These are crucial to determine the applicability of the proposed methodology to videos captured from the real world.
The good theoretical analysis of the geometrical issues invloved in the proposed model, need to be assessed with real data.

**Questions:**

Why didn't you extend the experiments with captured video data?

---

> ### Author Response · Authors · 2023-11-21
> **Response to Review 1rkU**
>
> We sincerely thank the reviewer for constructive comments and suggestions, which are very helpful for improving our paper. Please find our responses below. Revisions have also been made in the paper.
>
>
> **Q1. Consideration of real-world scenarios.**
>
> **Reply:** This is indeed an excellent comment. Although we have previously considered multiple test cases with respect to measurement noise and data missing due to visual obstruction, we further generated a video dataset simulating real-world scenarios to test the efficacy of our method. Here, we modeled the observed object as an ***irregular shape*** undergoing ***random self-rotational motion*** and ***size variations***, as shown in Appendix Figure S4a in the revised manuscript. Note that the size variations simulate changes in the camera's focal length when capturing the moving object in depth. The video frames were perturbed with a zero mean Gaussian ***noise*** (variance = 0.01). Moreover, a ***tree-like obstruction*** was introduced to further simulate the real-world complexity (e.g., the object might be obscured during motion) as depicted in Appendix Figure S4b. Despite these challenges, our method can discover the governing equations of the moving object in the reference coordinate system, showing its potential in practical applications under complex measurement conditions. Please refer to the added subsection of “Simulating Real-world Scenario” in the text (Page 8) and Appendix Section G (Pages 15 and 18) in the revised manuscript for more details. In addition, we also included a video for illustration in the supplementary material.
>
> We would like to bring the reviewer’s attention that, in the literature, limited efforts have been placed on directly distilling governing laws of dynamics directly from raw videos for moving targets in a 3D space, which represents a novel and interdisciplinary research domain. This challenge calls for a solution of fusing various techniques, including computer stereo vision, visual object tracking, and symbolic discovery of equations. To this end, we propose a vision-based approach to automatically uncover governing equations of nonlinear dynamics for 3D moving targets via raw videos recorded by a set of cameras. To the authors’ current knowledge, it might be ***the first attempt*** in the literature developing such an approach for discovery of nonlinear governing equations for 3D moving object, simply based on video measurements. Although testing the method on real-world measured videos lies in the pipeline of our research along this direction, our current efforts are placed on developing the computational framework. The effort of building a vision test setup in the authors’ lab is ongoing. We anticipate to complete the lab tests and obtain a set of real-world recorded videos to validate the proposed computational model in the near future.
>
> We hope the above reply and the revisions could clarify the reviewer’s concern. We would like to take this opportunity to thank the reviewer for the constructive comments. Please let us know if you have any other questions or comments. We look forward to hearing from you.

---

> ### Author Response · Authors · 2023-11-22
> **Looking forward to your feedback (Reviewer 1rkU)**
>
> Dear Reviewer 1rkU,
>
> We have posted the point-to-point reply to each question/comment raised by you and uploaded the revised version of our paper (with track changes marked in red). We believe your concerns have been fully addressed. Would you please let us know if you have any other questions?
>
> We look forward to your feedback. Thank you very much.
>
> Best regards,
>
> The Authors

---

### Official Review · Reviewer_cU4B · 2023-11-08

**Soundness:** 3 good
**Presentation:** 3 good
**Contribution:** 3 good
**Rating:** 8
**Confidence:** 3

**Summary:**

The authors propose a vision-based approach to automatically uncover governing equations of non linear dynamics for 3D moving targets via raw videos recorded by a set o fcameras.The proposed method consists of thee main steps namely 1) a target tracking module, 2) a Rodrigues’rotation formula-based coordinate transformation module, and 3) a spline-enhanced library-based sparse regressor.

**Strengths:**

1) The paper is well written and organized.
2) The proposed method is well designed and well detailed.
3) Experiments are well conducted and convincing.

**Weaknesses:**

The paper will be sometimes hard to read for a novice.

**Questions:**

None

---

> ### Author Response · Authors · 2023-11-21
> **Response to Review cU4B**
>
> We would like to sincerely thank the reviewer for the positive feedback.
>
> **Q1. Paper clarity.**
>
> **Reply:** Thanks for your comment. We have carefully revised our paper to improve the clarity. Here is a summary of our revision:
>
> - We have revised Figure 1 (e.g., notations and labeled modules) and placed it in front of the Methodology section, and also revised the Methodology part (particularly the subsections of Equation Discovery and Network training) in the text to improve the organization of the paper. We found some notation typos in the previous version, which have been fixed (see Figure 1 and Pages 5-6 in the revised manuscript). In addition, we removed the detailed formulations of the loss functions in Figure 1 and incorporated the corresponding notation description in the subsection of Network training (see Page 6 in the revised manuscript).
>
> - To further demonstrate the efficacy of our model, we generated a video dataset simulating real-world scenarios. Here, we modeled the observed object as an ***irregular shape*** undergoing ***random self-rotational motion*** and ***size variations***, as shown in Appendix Figure S4a in the revised manuscript. Note that the size variations simulate changes in the camera's focal length when capturing the moving object in depth. The video frames were perturbed with a zero mean Gaussian ***noise*** (variance = 0.01). Moreover, a ***tree-like obstruction*** was introduced to further simulate the real-world complexity (e.g., the object might be obscured during motion) as depicted in Appendix Figure S4b. Despite these challenges, our method can discover the governing equations of the moving object in the reference coordinate system, showing its potential in practical applications under complex measurement conditions. Please refer to the added subsection of “Simulating Real-world Scenario” in the text (Page 8) and Appendix Section G (Pages 15 and 18) in the revised manuscript for more details. In addition, we also included a video for illustration in the supplementary material.
>
> - We also carefully proofread our paper and fixed multiple typos.
>
> We hope the revisions could clarify the reviewer’s concern about the clarity of the paper.

---

### Official Review · Reviewer_4Dut · 2023-11-11

**Soundness:** 2 fair
**Presentation:** 2 fair
**Contribution:** 2 fair
**Rating:** 5
**Confidence:** 3

**Summary:**

The paper presented an approach for estimating the dynamics of a system using collected data from some observed solution. A system of three cameras was discussed for 3D point cloud estimation and object tracking for the purpose of collecting data about a system. Subsequently, the dynamics is estimated by fitting piece-wise polynomials to the time-series data. Method is evaluated using several synthetic datasets.

**Strengths:**

- The general problem was clearly discussed.
- Apart from some sections, generally, was easy to follow and understand the setup of the research questions.

**Weaknesses:**

- Relatively poor organization:
    - For example, some of the vital information are introduced early on, in Figure 1, and only referred and discussed in section 3.3. One has to referee to the formula in the figure to follow the discussion, although it was brief.
- Insufficient experimental results:
    - One of the main claimed contributions of the paper is the estimation of dynamics from data, collected using multiple cameras. None of the experimental results include such dataset. Infact all of the datasets were generated using a solution for some known dynamical systems. Please look at the questions section for more details.
- Contribution:
    - Contribution of the paper is not clear. Although it was stated to be the integration of different modules for tackling measurement noise and data gap, those points were not sufficiently argued for or validated with experimental results.

**Questions:**

- Data collection: The importance of Section 3.1. is not clear.
    - The relative camera positions are already assumed (section 3.3), however, it is stated that only one camera is calibrated (presumably known intrinsic camera parameter). Why is this assumption made?  is there any problem domain that forces this kind of assumption?
    - Why not use off the shelf 3D reconstruction (scene reconstruction) algorithms for the data collection together with scene-flow?

- More questions/suggestions:
    - Since the primarily claimed contribution of the work is using multi-camera systems for data collection, I suggest to work with real-world problems where different advantages and disadvantages of multi-camera-based data collection can be explored.
    - I understand, the common approach is to sample a particular solution and discover the dynamics from the data. In most cases, however, it seems the choice of test systems are chaotic which are exceptionally sensitive to initial condition/parameters. How robust are these discovery systems? since the data are samples from some solution which can be wildly different depending on the initial conditions.

---

> ### Author Response · Authors · 2023-11-21
> **Response to Review 4Dut (part 1)**
>
> We sincerely thank the reviewer for constructive comments and suggestions, which are very helpful for improving our paper. Please find our responses below. Revisions have also been made in the paper.
>
> **Q1. Paper organization.**
>
> **Reply:** Thanks for your comment. We have revised Figure 1 (e.g., notations and labeled modules) and placed it in front of the Methodology section. We also revised the Methodology part (particularly the subsections of Equation Discovery and Network training) in the text to improve the organization of the paper. Firstly, we found some notation typos in the previous version, which have been fixed (see Figure 1 and Pages 5-6 in the revised manuscript). Secondly, we removed the detailed formulations of the loss functions in Figure 1 and incorporated the corresponding notation description in the subsection of Network training (see Page 6 in the revised manuscript). Please also see below.
>
> *The loss function for this network comprises three main components, namely, the data component $L_d$, the physics component $L_p$, and the sparsity regularizer, given by:*
>
> $L(\mathbf{P}^*, \boldsymbol{\Lambda}^*, \Delta^*)= \arg \min _{\{\mathbf{P}, \boldsymbol{\Lambda}, \Delta\}}\left[L_d\left(\mathbf{P}, \Delta ; \mathcal{D}_r\right)+\alpha L_p\left(\mathbf{P}, \Delta, \boldsymbol{\Lambda} ; \mathcal{D}_c\right)\right]+\beta||\boldsymbol{\Lambda}||_0,$
>
> *where*
>
> $L_d= \frac{1}{N_m}\sum_{i=1}^{3} ||\mathbf{G}_m \mathbf{p}_i+\Delta_i-\mathbf{x}_i^m||_2^2,$
>
> $L_p= \frac{1}{N_c}\sum_{i=1}^{3} ||\boldsymbol{\Phi}(\mathbf{P}, \Delta) \boldsymbol{\lambda}_i-\dot{\mathbf{G}}^c \mathbf{p}_i||_2^2.$
>
> *Here, $\mathbf{G}_m$ denotes the spline basis matrix evaluated at the measured time instances, $\mathbf{x}_i^m$ the coordinates in each dimension after 3D reconstruction in the reference coordinate system (may be sparse or exhibit data gaps whereas $\dot{\mathbf{G}}_c$ the derivative of the spline basis matrix evaluated at the collocation instances. The term $\mathbf{G}_m \mathbf{p}_i$ is employed to fit the measured trajectory in each dimension, while $\dot{\mathbf{G}}^c \mathbf{p}_i$ is used to reconstruct the potential equations evaluated at the collocation instances. Additionally, $\mathbf{\Phi} \in \mathbb{R}^{N_c \times l}$ represents the collocation library matrix encompassing the collection of candidate terms, $||\boldsymbol{\Lambda}||_0$ the sparsity regularizer, $\alpha$ and $\beta$ the relative weighting parameters.*
>
>
> **Q2. Insufficient experimental results.**
>
> **Reply:** Thank you for this comment. There might be some misunderstanding about our discovery model. We would like to clarify that the discovery of the 3D dynamics was purely based on the generated video datasets (see the Data Generation in Section 4). After the dataset is generated, the governing equations of the dynamics is treated as **unknown**. The discovery was achieved following three subsequent steps: (1) We utilized the YOLO-v8 for object tracking in the recorded videos to obtain the pixel data in each video; (2) Leveraging Rodrigues' rotation formula and based on the learned calibrated camera, we reconstruct the 3D trajectory of the moving object; and (3) We employ the cubic B-splines to fit the trajectory and a library-based sparse regressor to uncover the underlying governing law of dynamics in the reference coordinate system. Hope this could clarify the reviewer’s concern.
>
>
> **Q3. Not clear contribution.**
>
> **Reply:** Thank you for this comment. Please note that, in the literature, limited efforts have been placed on directly distilling governing laws of dynamics directly from raw videos for moving targets in a 3D space, which represents a novel and interdisciplinary research domain. This challenge calls for a solution of fusing various techniques, including computer stereo vision, visual object tracking, and symbolic discovery of equations. To this end, we propose a vision-based approach to automatically uncover governing equations of nonlinear dynamics for 3D moving targets via raw videos recorded by a set of cameras. The approach is composed of three main blocks, which are uniquely positioned to address the aforementioned challenge. To the authors’ current knowledge, it might be ***the first attempt*** in the literature developing such an approach for discovery of nonlinear governing equations for 3D moving object, simply based on video measurements. Potential applications may include uncovering the law of dynamics of a flying object such as flock, drones, etc., for better understanding and perhaps controlling the behavior of the 3D object dynamics, simply based on raw videos.
>
> We would like to mention that this integrated framework excels in effectively addressing challenges associated with measurement noise and data gaps induced by imprecise target tracking. Results from extensive experiments demonstrate the efficacy of the proposed method. Hope this could clarify the reviewer’s concern on the contribution of this work.

---

> ### Author Response · Authors · 2023-11-21
> **Response to Review 4Dut (part 2)**
>
> **Q4. Camera calibration.**
>
> **Reply:** Thanks for this comment. Mapping image data to real-world scales requires calibrated cameras to accurately reflect the true scale. While having multiple calibrated cameras capturing images from different perspectives enhances the precision of reconstructing objects into 3D coordinates, we herein consider a challenging case where the information of only one calibrated camera is known *a priori*, **rather than the simple case** knowing the complete calibration information of multiple cameras. However, a single calibrated camera is insufficient for reconstructing the 3D trajectory coordinates of an object. To address this issue, we proposed a learning module to simultaneously calibrate other cameras and reconstruct the 3D trajectory, where the images captured by the calibrated camera serve as a reference. In particular, the uncalibrated cameras' parameters (e.g., the rotation angles of the image plane, the scaling factors) are learned by projecting images taken by other uncalibrated cameras onto the reference images through rigorous coordinate transformation. Such a learning module is vital to accurately distill the 3D trajectory of the moving object in a user-defined reference coordinate system.
>
>
> **Q5. Why not using off-the-shelf 3D reconstruction (scene reconstruction) algorithms.**
>
> **Reply:** Thank you for this comment. We did not employ off-the-shelf 3D reconstruction (scene reconstruction) algorithms for two primary reasons. Firstly, our approach is designed to be straightforward, and we only require the capture of object coordinates in the reference coordinate system instead of the entire 3D scene. Secondly, accurate extraction of the 3D motion trajectory of the object forms the foundation of our model for discovering the underlying governing equations. Our preliminary study showed that the off-the-shelf 3D scene reconstruction algorithms could not produce accurate reconstruction of the 3D trajectory of the moving target. Therefore, we designed the trajectory learning algorithm based on rigorous geometric transformation of coordinates and demonstrated its effectiveness over multiple examples.
>
>
> **Q6. Consideration of real-world scenarios.**
>
> **Reply:** This is indeed an excellent comment. Although we have previously considered multiple test cases with respect to measurement noise and data missing due to visual obstruction, we further generated a video dataset simulating real-world scenarios. Here, we modeled the observed object as an **irregular shape** undergoing **random self-rotational motion** and **size variations**, as shown in Appendix Figure S4a in the revised manuscript. Note that the size variations simulate changes in the camera's focal length when capturing the moving object in depth. The video frames were perturbed with a zero mean Gaussian **noise* (variance = 0.01). Moreover, a **tree-like obstruction** was introduced to further simulate the real-world complexity (e.g., the object might be obscured during motion) as depicted in Appendix Figure S4b. Despite these challenges, our method can discover the governing equations of the moving object in the reference coordinate system, showing its potential in practical applications under complex measurement conditions. Please refer to the added subsection of “Simulating Real-world Scenario” in the text (Page 8) and Appendix Section G (Pages 15 and 18) in the revised manuscript for more details. In addition, we also included a video for illustration in the supplementary material.
>
> It is noted that since video cameras offer a direct and less expensive means to measure flying objects, discovering the nonlinear dynamics of those systems (e.g., flock, unidentified drones) which are intractable to measure becomes possible. This obviously forms the advantage of using videos for discovery of nonlinear dynamics for 3D moving target. However, this calls for sophisticated algorithms to tackle this challenge, fusing various techniques including computer stereo vision, visual object tracking, and symbolic discovery of equations.
>
>
> **Q7. Model robustness with respect to the initial conditions.**
>
> **Reply:** The reviewer is right if we solve the equations (e.g., using a time integration method) of a chaotic system, the solution will be sensitive to the initial conditions. However, in the context of equation discovery, we formed an optimization problem minimizing the sum of the data discrepancy and the residual of the underlying equation, instead of solving the equation, essentially alleviating the issue of sensitivity to initial conditions. This common practice has been proven effective in the discovery of governing equations even for chaotic dynamics in the literature.
>
> We hope the above replies and the revisions could clarify the reviewer’s concern. We would like to take this opportunity to thank the reviewer for the detailed and constructive comments. Please let us know if you have any other questions or comments.

---

> ### Author Response · Authors · 2023-11-22
> **Looking forward to your feedback (Reviewer 4Dut)**
>
> Dear Reviewer 4Dut,
>
> We have posted the point-to-point reply to each question/comment raised by you and uploaded the revised version of our paper (with track changes marked in red). We believe your concerns have been fully addressed. Would you please let us know if you have any other questions?
>
> We look forward to your feedback. Thank you very much.
>
> Best regards,
>
> The Authors

---

### Official Review · Reviewer_NGgK · 2023-11-12

**Soundness:** 2 fair
**Presentation:** 2 fair
**Contribution:** 1 poor
**Rating:** 5
**Confidence:** 3

**Summary:**

The paper describes an approach to discover the equations governign the non-linear dynamics of a moving target and claims one of the first approach to do it on collection of 2D videos. Synchronised cameras proposed used to capture object motion and track it in 3D. A spline model aims to denoise the object motion and fit this to sparse coefficients to some fixed candidate functions given in a library.

**Strengths:**

The paper boast of proposing first approach to discover the equations of dynamics for moving target while using the 2D data. The presented results demonstrate superior performance of the approach when compared to the baseline. While the paper lacks crucial information and is technically dense, it is generally well written.

**Weaknesses:**

- The paper rely to a great extent on figure 1 but the proposed methodology has not been explained well either in the figure or in the text. A lot of focus has been given to finding out the noisy 3D tracks from calibrated camera rigs - where I don't see any novelty. Vet little any attention has been given to actual spline driven fitting of reconstructed track and learning the equation parameters. I find that many of the notations in figure 1 still remain undescribed and the paper jumps from writing about YoLo V8 (perhaps for object detection), "learning" 3D trajectory (which looks like reconstructing 3D from 2D detections) to equation fitting which is regression. Are these three different stages of proposed pipeline of everything is learned end to end?

- The paper boasts heavily of being the first one to find these dynamics from images. All presented results however are on synthetic data and does not really justify an expansive 3 camera setup to track and 3D reconstruct the object locations. Authors did not explain well what the type of objects one might want to track, and the challenges associated with the same. Tracking square and star projections using three cameras and YoLo8 seems an overkill and even if one can do so what is novel in doing so?
I think the paper could do with devoting more time in explaining why the discovery of dynamics is important and how the proposed setup helps in tracking the real objects better for the same. I understand the challenges in getting the real data for the problem at hand but think it is extremely important to present with some real results - even qualitative results with 2D/3D tracking could work leaving the equation discovery out of evaluation.

**Questions:**

- looking at table S2, it seems that the problem of equation discovery seems to be an ill posed problem and require a very densely sampled set of observations. Can author comment on the terms used in the library and the interdependency in those terms to model a sparse set of trajectories?

- Shouldn't you also present the trajectory noise as an independent measure of good 3D tracking to isolate errors in 3D tracking and equation discovery?

---

> ### Author Response · Authors · 2023-11-21
> **Response to Review NGgK (Part 1)**
>
> We sincerely thank the reviewer for the constructive comments and suggestions, which are very helpful for improving our paper. The revisions have been incorporated in the revised manuscript marked in red.
>
> **Q1. Methodology not explained well.**
>
> **Reply:** Thanks for the reviewer’s great comment. We have revised Figure 1 and the Methodology part (particularly the subsections of Equation Discovery and Network training) in the text to improve the clarity of the paper. Firstly, we found some notation typos in the previous version, which have been fixed (see Figure 1 and Pages 5-6 in the revised manuscript). Secondly, we removed the detailed formulations of the loss functions in Figure 1 and incorporated the corresponding notation description in the subsection of Network training (see Page 6 in the revised manuscript). Please also see below.
>
> *The loss function for this network comprises three main components, namely, the data component $L_d$, the physics component $L_p$, and the sparsity regularizer, given by:*
>
> $L(\mathbf{P}^*, \boldsymbol{\Lambda}^*, \Delta^*)= \arg \min _{\{\mathbf{P}, \boldsymbol{\Lambda}, \Delta\}}\left[L_d\left(\mathbf{P}, \Delta ; \mathcal{D}_r\right)+\alpha L_p\left(\mathbf{P}, \Delta, \boldsymbol{\Lambda} ; \mathcal{D}_c\right)\right]+\beta||\boldsymbol{\Lambda}||_0,$
>
> *where*
>
> $L_d= \frac{1}{N_m}\sum_{i=1}^{3} ||\mathbf{G}_m \mathbf{p}_i+\Delta_i-\mathbf{x}_i^m||_2^2,$
>
> $L_p= \frac{1}{N_c}\sum_{i=1}^{3} ||\boldsymbol{\Phi}(\mathbf{P}, \Delta) \boldsymbol{\lambda}_i-\dot{\mathbf{G}}^c \mathbf{p}_i||_2^2.$
>
> *Here, $\mathbf{G}_m$ denotes the spline basis matrix evaluated at the measured time instances, $\mathbf{x}_i^m$ the coordinates in each dimension after 3D reconstruction in the reference coordinate system (may be sparse or exhibit data gaps whereas $\dot{\mathbf{G}}_c$ the derivative of the spline basis matrix evaluated at the collocation instances. The term $\mathbf{G}_m \mathbf{p}_i$ is employed to fit the measured trajectory in each dimension, while $\dot{\mathbf{G}}^c \mathbf{p}_i$ is used to reconstruct the potential equations evaluated at the collocation instances. Additionally, $\mathbf{\Phi} \in \mathbb{R}^{N_c \times l}$ represents the collocation library matrix encompassing the collection of candidate terms, $||\boldsymbol{\Lambda}||_0$ the sparsity regularizer, $\alpha$ and $\beta$ the relative weighting parameters.*
>
>
> **Q2: Novelty of the 3D trajectory learning module.**
>
> **Reply:** Thanks for this question. Accurate extraction of the 3D motion trajectory of the object forms the foundation of our model for discovering the underlying governing equations. We herein consider a challenging case where only one camera is calibrated as *a priori* knowledge, **rather than the simple case** knowing the complete calibration information of all the three cameras. However, a single calibrated camera is insufficient for reconstructing the 3D trajectory coordinates of an object. To address this issue, we proposed a learning module to simultaneously calibrate other cameras and reconstruct the 3D trajectory, where the images captured by the calibrated camera serve as a reference. In particular, the uncalibrated cameras' parameters (e.g., the rotation angles of the image plane, the scaling factors) are learned by projecting images taken by other uncalibrated cameras onto the reference images through rigorous coordinate transformation. Such a learning module is vital to accurately distill the 3D trajectory of the moving object in a user-defined reference coordinate system. Hope this clarifies the reviewer’s concern.

---

> ### Author Response · Authors · 2023-11-21
> **Response to Review NGgK (Part 2)**
>
> **Q3: Adding description of the spline fitting and equation parameter learning.**
>
> **Reply:** Thanks for your suggestion. Indeed, the spline fitting and equation parameter learning are performed simultaneously. Since the regularizer $||\boldsymbol{\Lambda}||_0$ leads to an NP-hard optimization issue, we apply an Alternate Direction Optimization (ADO) strategy to optimize the total loss function (see our reply to Q1). The interplay of spline interpolation and sparse equations yields subsequent effects: the spline interpolation ensures accurate modeling of the system's response, its derivatives, and the candidate function terms, thereby laying the foundation for constructing the governing equations. Simultaneously, the equations represented in a sparse manner synergistically constrain spline interpolation and facilitate the projection of accurate candidate functions. We have added a section in the Appendix (e.g., Section C on Pages 14-15) to describe the ADO algorithm. Please also see below.
>
> *Addressing the optimization problem directly via gradient descent is highly challenging, given the NP-hard problem induced by the $\ell_0$ regularizer. Alternatively, relaxing $\ell_0$ to $\ell_1$ eases the optimization process but only provides a loose promotion of sparsity. We employ an alternating direction optimization (ADO) strategy that hybridizes gradient descent optimization and sparse regression. The approach involves decomposing the overall optimization problem into a set of tractable sub-optimization problems, formulated as follows:*
>
> $\boldsymbol{\lambda}_{i}^{(k+1)}:=\arg\min _{\boldsymbol{\lambda}_i} ||\boldsymbol{\Phi} (\mathbf{P}^{(k)},\Delta^{(k)})\boldsymbol{\lambda}_i-\dot{\mathbf{G}}^c\mathbf{p}_i^{(k)}||_2^2 + \beta||\boldsymbol{\lambda}_i||_0,$
>
> $\\{\mathbf{P}^{(k+1)}, \tilde{\boldsymbol{\Lambda}}^{(k+1)}, {\Delta}^{(k+1)}\\}:=\arg \min _{\\{\mathbf{P}, \tilde{\boldsymbol{\Lambda}}, {\Delta} \\}} [\mathcal{L}_d(\mathbf{P}, \Delta)+\alpha \mathcal{L}_p(\mathbf{P}, \tilde{\boldsymbol{\Lambda}})],$
>
> *where $k$ denotes the index of the alternating iteration, $\tilde{\boldsymbol{\Lambda}}$ consists of only the non-zero coefficients in $\tilde{\boldsymbol{\Lambda}}^{(k+1)} = \\{\boldsymbol{\lambda}_1^{(k+1)}, \boldsymbol{\lambda}_2^{(k+1)}, \boldsymbol{\lambda}_3^{(k+1)}\\}$.
> In each iteration, $\boldsymbol{\Lambda}^{(k+1)}$ (shown in the first equation) is determined using the STRidge algorithm with adaptive hard thresholding, pruning small values by assigning zero. The optimization problem in the second equation can be solved via gradient descent to obtain the updated $\mathbf{P}^{(k+1)}, \tilde{\boldsymbol{\Lambda}}^{(k+1)}$ and ${\Delta}^{(k+1)}$ with the remaining terms in $\boldsymbol{\Phi}$ (e.g., redundant terms are pruned). This iterative process continues for multiple iterations until achieving a final balance between the spline interpolation and the pruned equations.*
>
>
> **Q4: The pipeline of the proposed model and the importance of the work.**
>
> **Reply:** Thanks for this question. The role of the object tracking module is to leverage target recognition capabilities to extract the pixel positions of moving objects in the image plane of each video. Subsequently, the coordinate transformation module is employed to learn the parameters of other uncalibrated cameras. This enables the reconstruction of coordinates in the reference coordinate system. Finally, a spline-enhanced library-based sparse regressor is employed to uncover the governing equations based on the extracted 3D trajectory data. Each of the three modules is end-to-end learning.
>
> Please note that limited efforts have been placed on directly distilling governing laws of dynamics directly from raw videos for moving targets in a 3D space, which represents a novel and interdisciplinary research domain. This challenge calls for a solution of fusing various techniques, including computer stereo vision, visual object tracking, and symbolic discovery of equations. We aimed to tackle this challenge by proposing an effective learning approach. Potential applications may include uncovering the law of dynamics of a flying object such as flock, drones, etc., for better understanding and perhaps controlling the behavior of the 3D object dynamics, simply based on videos.

---

> ### Author Response · Authors · 2023-11-21
> **Response to Review NGgK (Part 3)**
>
> **Q5: Consideration of more complex datasets and real-world scenarios.**
>
> **Reply:** This is indeed an excellent comment. Although we have previously considered multiple cases (e.g., measurement noise, data missing due to visual obstruction), we followed the reviewer’s great suggestion and further generated a video dataset simulating real-world scenarios. Here, we modeled the observed object as an ***irregular shape*** undergoing ***random self-rotational motion*** and ***size variations***, as shown in Appendix Figure S4a in the revised manuscript. Note that the size variations simulate changes in the camera's focal length when capturing the moving object in depth. The video frames were perturbed with a zero mean Gaussian ***noise*** (variance = 0.01). Moreover, a ***tree-like obstruction*** was introduced to further simulate the real-world complexity (e.g., the object might be obscured during motion) as depicted in Appendix Figure S4b. Despite these challenges, our method can discover the governing equations of the moving object in the reference coordinate system, showing its potential in practical applications under complex measurement conditions.
>
> Please refer to the added subsection of “Simulating Real-world Scenario” in the text (Page 8) and Appendix Section G (Pages 15 and 18) in the revised manuscript for more details. In addition, we also included a video for illustration in the supplementary material.
>
>
> **Q6: Ill-posed problem of equation discovery and very densely sampled set of observations.**
>
> **Reply:** Thank you for this comment. Firstly, the library may consist of constant, polynomial, and trigonometric terms. The greater the variety of terms in the function library, the higher likelihood that the library encompasses correct terms. Insufficient variety of terms may result in overlooking genuine terms, leading to an inability to identify the correct expression for the equation. In this paper, the library of candidate functions includes combinations of system states with polynomials up to the third order following the practice used in Brunton et al. [113(15):3932–3937, 2016].
>
> In the experiments presented in Table S2, the equations are discovered based on the 3D coordinate trajectories reconstructed from videos within the reference coordinate system. Due to visual obstruction of the moving object in the video, the reconstructed trajectory might have data missing and gaps. As a result, the set of observations are sparse, instead of densely sampled. This has been discussed in the subsection of “Random Block Missing Data Effect”. In addition, we have added Figure S3 in the Appendix (see page 15) of the revised paper to illustrate this aspect.
>
>
> **Q7. Trajectory noise effect.**
>
> **Reply:** Thank you for your comment. Actually, we have conducted experiments under the same condition of 3D tracking and trajectory reconstruction, considering different levels of measurement noise. We systematically evaluated the performance of our method under the influence of trajectory noise using both qualitative and quantitative metrics. The assessment is detailed in the subsection of “Noise Effect” and Figure 3 in the paper (see Page 8).
>
> We hope the above replies and the revisions could clarify the reviewer’s concern. We would like to take this opportunity to thank the reviewer for the detailed and constructive comments. Please let us know if you have any other questions or comments. We look forward to hearing from you.

---

> ### Author Response · Authors · 2023-11-22
> **Looking forward to your feedback (Reviewer NGgK)**
>
> Dear Reviewer NGgK,
>
> We have posted the point-to-point reply to each question/comment raised by you and uploaded the revised version of our paper (with track changes marked in red). We believe your concerns have been fully addressed. Would you please let us know if you have any other questions?
>
> We look forward to your feedback. Thank you very much.
>
> Best regards,
>
> The Authors

---

### Author Response · Authors · 2023-11-22
**Revised paper uploaded and response to reviewers’ comments posted**

Dear Reviewers and AC:

Firstly, we would like to thank the reviewers for the constructive comments, which are very helpful for improving our paper. We have posted the point-to-point reply to each question/comment raised by you and uploaded the revised version of our paper (with track changes marked in red). We believe the reviewers' concerns have been fully addressed. Here is a summary listing the main revisions included in the paper:

- **Paper clarity:** We have revised Figure 1 and the Methodology part (particularly the subsections of Equation Discovery and Network Training) in the text to improve the clarity and organization of the paper. We have also carefully proofread the paper and fixed multiple typos and notation inconsistencies. Please see Pages 3, 5 and 6 as well as the Appendix Section G (Pages 14-15) in the revised manuscript for more details.

- **Real-world dataset:** Although we have previously considered multiple test cases with respect to measurement noise and data missing due to visual obstruction, we followed the suggestion by Reviewers NGgK, 4Dut and 1rkU, and further generated a video dataset simulating real-world scenarios to test the efficacy of our method. Here, we modeled the observed object as an ***irregular shape*** undergoing ***random self-rotational motion*** and ***size variations***, as shown in Appendix Figure S4a in the revised manuscript. Note that the size variations simulate changes in the camera's focal length when capturing the moving object in depth. The video frames were perturbed with a zero mean Gaussian ***noise*** (variance = 0.01). Moreover, a ***tree-like obstruction*** was introduced to further simulate the real-world complexity (e.g., the object might be obscured during motion) as depicted in Appendix Figure S4b. Despite these challenges, our method can discover the governing equations of the moving object in the reference coordinate system, showing its potential in practical applications under complex measurement conditions. Please refer to the added subsection of “Simulating Real-world Scenario” in the text (Page 8) and Appendix Section G (Pages 15 and 18) in the revised manuscript for more details. In addition, we also included a video for illustration in the supplementary material.

We would like to mention that, although testing the method on real-world measured videos lies in the pipeline of our research along this direction, our current efforts are placed on developing the computational framework for distilling governing laws of dynamics directly from raw videos for moving targets in a 3D space. The effort of building a vision test setup in the authors’ lab is ongoing. We anticipate completing the lab tests and obtaining a set of real-world recorded videos to validate the proposed computational model in the near future.

Please do feel free to let us know if you have any further questions. Thank you very much.

Best regards,

The Authors